# Multivariate analysis of disorder in metal–organic frameworks

Adam F. Sapnik [1], Irene Bechis [2], Alice M. Bumstead [1], Timothy Johnson [3], Philip A. Chater [4], David A. Keen [5], Kim E. Jelfs [2] & Thomas D. Bennett [1✉]

The rational design of disordered frameworks is an appealing route to target functional materials. However, intentional realisation of such materials relies on our ability to readily characterise and quantify structural disorder. Here, we use multivariate analysis of pair distribution functions to fingerprint and quantify the disorder within a series of compositionally identical metal–organic frameworks, possessing different crystalline, disordered, and amorphous structures. We find this approach can provide powerful insight into the kinetics and mechanism of structural collapse that links these materials. Our methodology is also extended to a very different system, namely the melting of a zeolitic imidazolate framework, to demonstrate the potential generality of this approach across many areas of disordered structural chemistry.

[1] Department of Materials Science and Metallurgy, University of Cambridge, Cambridge CB3 0FS, UK. [2] Department of Chemistry, Imperial College London, Molecular Sciences Research Hub, White City Campus, London W12 0BZ, UK. [3] Johnson Matthey Technology Centre, Blount's Court, Sonning Common, Reading RG4 9NH, UK. [4] Diamond Light Source Ltd, Diamond House, Harwell Campus, Didcot, Oxfordshire OX11 0DE, UK. [5] ISIS Neutron and Muon Facility, Rutherford Appleton Laboratory, Harwell Campus, Didcot, Oxfordshire OX11 0QX, UK. ✉email: tdb35@cam.ac.uk

Disorder is emerging as an increasingly prevalent and powerful means to drive functionality within framework materials[1]. Once regarded as 'perfect' crystalline structures, they have been found to harbour a variety of types of disorder[2–5]. Even frameworks ostensibly believed to be highly symmetric and rigid, such as UiO-66, have been found to possess some degree of disorder[6,7]. Consequently, disorder has become pervasive amongst this class of materials and comes in many guises, including static, dynamic and topological, random or correlated, unexpected or intentionally engineered[8,9].

Characterising disorder is challenging; detecting the presence of defects or determining the structure of an amorphous framework, for example, are both non-trivial endeavours[10–13]. However, such knowledge is crucial to truly take advantage of the functionality that disorder can impart. Unfortunately, the use of traditional crystallographic methods—sensitive only to the average structure—are largely blind to structural disorder[14]. Instead, local structure techniques such as nuclear magnetic resonance spectroscopy, X-ray absorption spectroscopy and electron microscopy are commonly employed to probe these challenging structures[15–17].

Total scattering and pair distribution function (PDF) analysis has found itself at the forefront of this area of structural chemistry[18–21]. Total scattering data, which incorporates both Bragg and diffuse scattering, are obtained from a diffraction experiment and has been used extensively to probe the intermediate and extended length scales of order within inorganic glasses such as Si and $GeO_2$[22–25]. By making use of the Fourier transform of the total scattering, the PDF is obtained, which is a real-space mapping of the two-body interactions and is sensitive to both the local and extended structure[26]. Previously, the collection of total scattering data represented a significant bottleneck in performing PDF studies, requiring long measurement times[18]. However, with the advent of synchrotron radiation sources and rapid area detectors, high-quality total scattering data can now be obtained within a matter of seconds[27]. PDF analysis has been used to provide insight into the negative thermal expansion behaviour in $ZrW_2O_8$, the structure of metal oxide nanoparticles, in-situ formation of hybrid materials, operando studies on battery materials and, of course, studies on structural disorder[28–36].

From a practical perspective, advances in instrumentation mean the bottleneck now lies in data analysis. While many approaches toward PDF analysis exist, such as small- and large-box modelling, differential analysis and even machine learning, it remains difficult to readily characterise the processes by which disorder is often incorporated within framework materials, such as structural collapse or melting[18,37–42]. This challenge of interpreting complex data is certainly not a new one and extends across a breadth of fields including chemistry, medicine, and biology[43]. Multivariate analysis seeks to simplify this challenge by reducing complex data into its latent components.

Two prominent methods of multivariate analysis include principal component analysis (PCA) and non-negative matrix factorisation (NMF)[44–47]. Both techniques take $m$ datasets and attempt to describe them in terms of linear combinations of $n$ components ($n < m$). In PCA, these components are constrained to be orthogonal functions, whereas in NMF they are constrained to be positive functions. More recently, hybrid methods combining NMF with the Metropolis Monte Carlo algorithm, known as MMF, have been employed to overcome the convex nature of NMF's minimisation problem[48]. The PCA procedure projects the original dataset onto the eigenvectors of an orthogonal basis set whose directions are found by simultaneously maximizing the variance of the data and minimising the projection error. The corresponding eigenvalues of these components represent their relative importance. The first principal component has the largest eigenvalue and accounts for the largest proportion of the total variance in the data, the second principal component for the next largest proportion of the variance and so on.

When applied to PDFs the principal components can encode atom–atom correlations, distortions or noise within the data[49]. Components that resemble the typical form of a PDF represent atom–atom correlations within the dataset. For PDFs obtained from X-ray scattering data, this takes the form of positive peaks above a $-4\pi\rho r$ baseline (which is most obvious at low-$r$, $\rho$ is the sample density) and an apparent envelope to the PDF peaks that decreases at high-$r$[18]. For neutron scattering data, peaks may have negative intensity below the $-4\pi\rho r$ baseline due to the negative scattering cross-section of certain isotopes. Components that deviate from the typical form of a PDF describe structural distortions of the atomic structure. While components that are dominated by high-frequency signals typically represent noise in the data. Components obtained through PCA are subject to certain physical expectations to ensure they are meaningful, as set out by Chapman et al.[49].

Multivariate analysis has provided meaningful insights in PDF studies across the field of materials chemistry. Key examples have been reported by Chapman et al., including solvation shells of metal salt aqueous solutions, formation of zeolite-supported Ag nanoparticles, and electrochemical lithiation of $RuO_2$[49]. Subsequent studies have included the cycling of lithium-ion batteries, the interface between Fe and $Fe_3O_4$ and pharmaceutical amorphous solid dispersions[33,34,48,50]. In the majority of cases, PDF studies coupled with PCA focus on phase mixtures and it is principal components that contain atom–atom correlations that have received the most attention. Studies investigating components that reveal structural distortions are less prevalent due to their more challenging interpretation.

Disordered materials are often obtained through the structural distortion of a crystalline material. For example, the collapse of a framework under the application of uniaxial or hydrostatic compression[51]. The structural distortions that occur to generate these disordered states are complex to study, yet the PDF will be sensitive to these changes in the atomic structure. Principal components that describe structural distortions will detect these subtle modulations of the PDF as the structure changes. By virtue of this, details of the structural distortion that has occurred can be inferred from distortion components. As a trivial example, consider an A–X bond of length $r$ that increases by an amount $\delta r$ (Fig. 1a, b). The peak within the PDF corresponding to the A–X interaction will exhibit a shift in position by $\delta r$ to higher $r$, which will be captured as a unique signature within a distortion component (Fig. 1c, d). Common changes that are observed in the PDF include peak broadening and peak shifts, which both give rise to distinctive features in the distortion (Supplementary Figs. 1, 2). In reality, the atomic structure and structural distortions in framework materials will be far more complex, involving secondary building units with various routes of deformation, but will nonetheless lead to a unique distortion.

We focus on four compositionally identical metal–organic frameworks (MOFs), the crystalline MIL-100 framework, the topologically disordered Fe-BTC framework and their two amorphous counterparts $a_m$MIL-100 and $a_m$Fe-BTC ($a_m$ denotes amorphisation *via* ball milling, in these examples for 30 minutes), respectively. We have previously studied these materials in detail in Refs. [13,35]. The structural chemistry of MIL-100 and Fe-BTC consists of oxo-centred trimer units formed from three Fe (III) ions (Fig. 1e), with four trimer units linked by 1,3,5-benzentricarboxylate anions in tetrahedral assemblies (Fig. 1f). These tetrahedral assemblies are linked periodically in MIL-100 and in a disordered network within Fe-BTC. Ball milling of MIL-100 and Fe-BTC results in the breaking of metal–linker bonds,

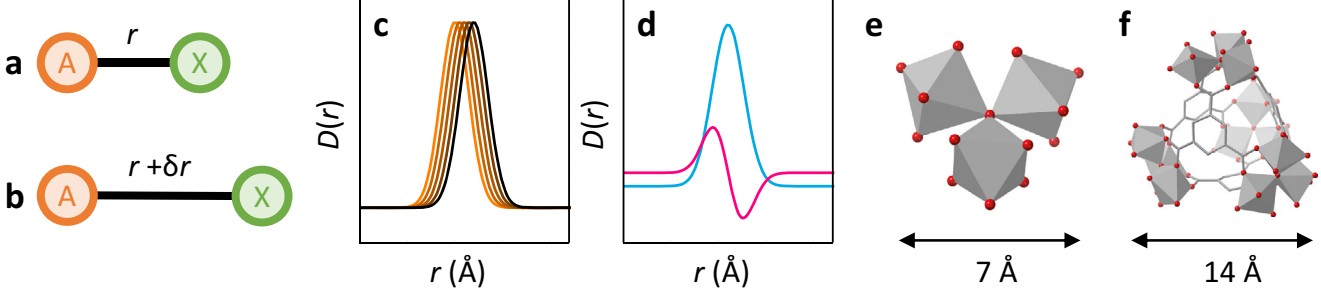

**Fig. 1 Distortion components and structural chemistry. a** Illustrative A–X bond of length $r$. **b** The A–X bond following a structural distortion which increases its length by $\delta r$. **c** PDFs tracking the A–X bond between its original (orange) and distorted (black) state. **d** The first (blue) and second (pink) principal components. The latter acts as a signature of the distortion that occurred to the A–X bond. The x-axis scale is the same in (**c**) and (**d**). Structure of (**e**) oxo-centred trimer unit and (**f**) tetrahedral assembly. $FeO_6$ octahedra (grey), O (red), C (grey) and H omitted for clarity.

incorporating structural disorder, to form their amorphous counterparts (Fig. 2a)[13,16,35,52,53].

In this study, we explore how PCA can be applied to characterise and quantify disorder in the MIL-100 and Fe-BTC family, with a particular focus on components that encode distortions in the structure, which have, to date, received less attention in PCA studies. By studying the distortions, we can directly fingerprint the motifs of structural disorder. In turn, we compare the structures of MIL-100 and Fe-BTC and their amorphous counterparts. We demonstrate how PCA can be used to gain insight into the mechanism and kinetics of structural collapse in MIL-100 and Fe-BTC. Finally, we extend our discussion to a system with very different structural chemistry, namely the melting of a zeolitic imidazolate framework, demonstrating the potential generality of this approach to a broad spectrum of disordered structures.

## Results

**Powder X-ray diffraction.** Following the synthesis of the four materials, preliminary investigations were carried out using powder X-ray diffraction (Fig. 2b). The diffraction pattern for MIL-100 contained sharp Bragg scattering and was confirmed to be phase pure through Pawley refinement of the data (Supplementary Fig. 3 and Supplementary Table 1). Fe-BTC possessed neither sharp Bragg nor broad diffuse scattering. Instead, characteristic regions of weak scattering were observed that were centred around the clusters of intense Bragg scattering in MIL-100. $a_m$MIL-100 and $a_m$Fe-BTC appeared amorphous, displaying characteristic 'halos' of diffuse scattering akin to that observed in glassy MOF materials[54].

Both $a_m$MIL-100 and $a_m$Fe-BTC exhibited significantly broader and lower intensity features than their parent materials, suggesting a large degree of structural disordering occurred during ball milling. It has previously been confirmed that the reduction in particle size upon milling is minimal within these materials[35]. Upon closer inspection of the diffuse scattering, very subtle differences were apparent (Fig. 2b **inset**). Most notably, at low scattering angle (<15°) the intensity in $a_m$Fe-BTC was slightly greater and marginally narrower than for $a_m$MIL-100. Whilst certainly not definitive, these variations may point towards subtle differences between the structures of $a_m$MIL-100 and $a_m$Fe-BTC.

**X-ray total scattering.** To further probe the structures of the four materials, synchrotron X-ray total scattering data were collected using the I15-1 beamline at the Diamond Light Source (Oxfordshire, UK). The data were, in general, similar to the lab-source diffraction data, but with a superior $Q_{max}$ afforded by the high-energy radiation (Fig. 3a). As expected, the X-ray total scattering displayed (i) Bragg scattering for MIL-100, (ii) broad

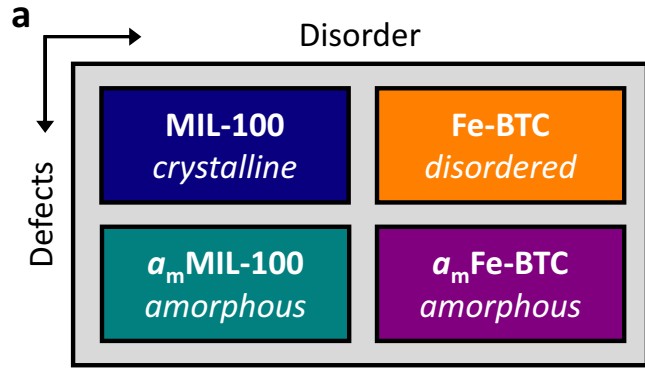

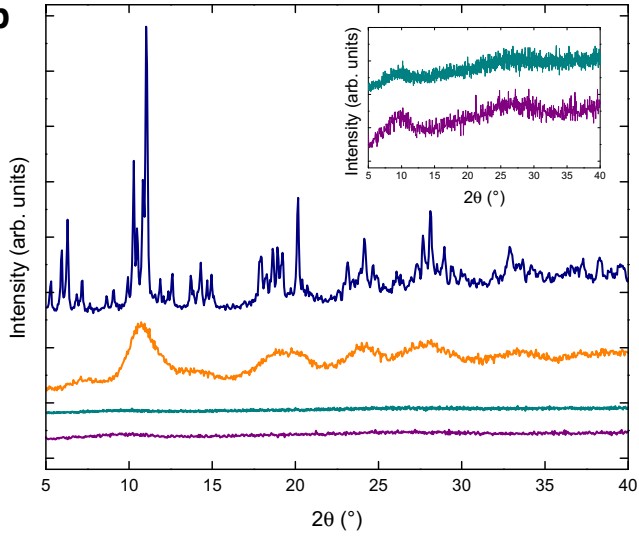

**Fig. 2 Structural characterisation. a** Diagram relating the four materials of interest. **b** Powder X-ray diffraction data for MIL-100 (navy), Fe-BTC (orange), $a_m$MIL-100 (turquoise) and $a_m$Fe-BTC (purple). Inset shows both amorphous materials. Data for MIL-100, Fe-BTC and $a_m$MIL-100 reproduced from Refs. [13,35]. Data offset for clarity.

diffuse scattering that was largely featureless for $a_m$MIL-100 and $a_m$Fe-BTC, and (iii) scattering intermediate to the crystalline and amorphous materials for Fe-BTC. It is worth also highlighting the reduced reciprocal space resolution of these data compared to the lab-source powder diffraction data. For example, the single peak observed for MIL-100 in the total scattering data at $Q = 0.75\,\text{Å}^{-1}$ corresponds to the cluster of Bragg peaks around 10° in the powder diffraction data, demonstrating considerable instrumental broadening. However, the data for Fe-BTC, $a_m$MIL-100 and

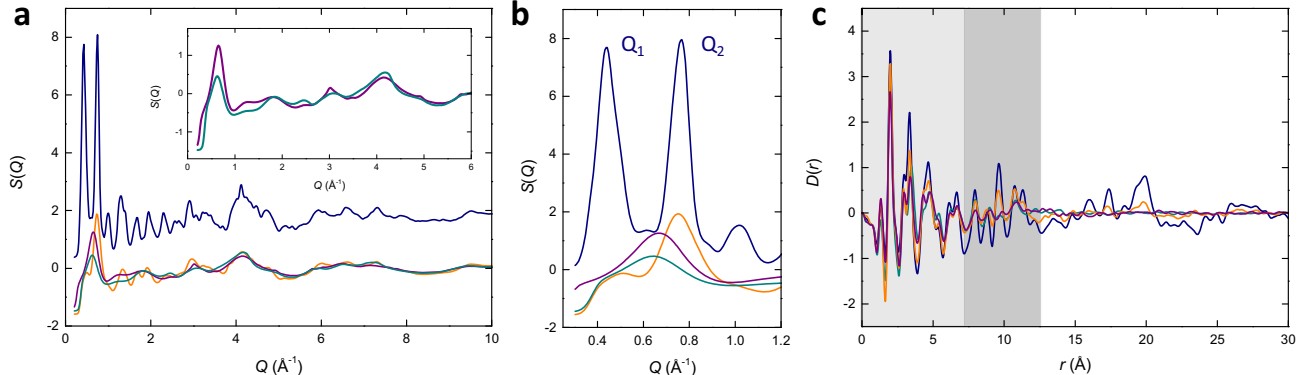

**Fig. 3 Reciprocal and real space structure. a** X-ray total scattering for MIL-100 (navy), Fe-BTC (orange), $a_m$MIL-100 (turquoise) and $a_m$Fe-BTC (purple). MIL-100 offset for clarity. Inset highlights $a_m$MIL-100 and $a_m$Fe-BTC. **b** Comparison of the low-$Q$ structure factor data. **c** X-ray pair distribution functions for MIL-100 (navy), Fe-BTC (orange), $a_m$MIL-100 (turquoise) and $a_m$Fe-BTC (purple). The light grey region highlights peaks largely originating from the trimer unit and dark grey those from the tetrahedral assemblies. Data for MIL-100, Fe-BTC and $a_m$MIL-100 from Refs. [13,35].

$a_m$Fe-BTC were appreciably broad when measured by either instrument. Therefore, we attribute the broadening of these three samples in the total scattering data to inherent sample contributions, while the data for MIL-100 are broadened predominantly due to the limited instrumental resolution.

Some of the largest variations in the structure factors were observed in the low-$Q$ region (Fig. 3b). Here, MIL-100 exhibited two regions of intense scattering that we denote $Q_1$ (0.42 Å$^{-1}$) and $Q_2$ (0.75 Å$^{-1}$). Crystallographically, these peaks are dominated by the (422) and (662) Bragg peaks, respectively. Given the extremely large unit cell of MIL-100, these crystallographic directions do not provide a particularly intuitive basis to interpret the structure factors. Equally, $Q_1$ and $Q_2$ can be compared to the partial structure factors, $A_{ij}(Q)$ as defined in Ref. [55], to investigate the pairwise atomic contributions to these peaks (Supplementary Fig. 4). Unsurprisingly, this revealed that the majority of the $A_{ij}(Q)$ functions contribute in some way to $Q_1$ and $Q_2$. The dominant contributions to $Q_1$ and $Q_2$ were C–O/O–Fe and O–Fe/Fe–Fe/O–O, respectively. The calculated structure factor also revealed a peak around $Q = 0.23$ Å$^{-1}$ which was not experimentally observed due to the design of the beamline.

Another, perhaps more intuitive, means of interpreting $Q_1$ and $Q_2$ lies in their real-space, Fourier components that describe the electron density fluctuations within MIL-100—an approach often employed in the study of inorganic glasses[22–25]. The localised nature of $Q_1$ and $Q_2$ in reciprocal space relate to delocalised features in real space; the position of the peak denotes the period of the oscillation, and the sample contribution to the peak width describes the correlation length over which they persist. $Q_1$ describes an electron density fluctuation with a period of approximately 15.0 Å and $Q_2$ a period of 8.4 Å, predominantly in the directions of the (422) and (662) lattice vectors, respectively. Given the limited reciprocal space resolution of the total scattering data, we note that the correlation lengths associated with $Q_1$ and $Q_2$ in the total scattering are dictated by the instrumental broadening and not the sample itself. In a crystalline material, the expected narrow width of the Bragg peaks implies strong structural coherence is present. In other words, separated regions of the crystal are in register with each other, even over large distances. Within disordered materials, this coherence is much shorter, due to different regions of their structure being out of register with each other, contributing, in part, to the broadening of the peaks.

Relating these electron density fluctuations to the hierarchical structure of MIL-100, it becomes apparent that the 8.4 Å ($Q_2$) oscillation qualitatively arises from the trimer unit and the 15.0 Å

($Q_1$) oscillation from the tetrahedral assemblies (and noting that the trimer unit contains the highest electron density in the material) (Fig. 1e, f). The spatial persistence of these building blocks throughout the periodic framework results in a continuous variation in electron density with periods described by $Q_1$ and $Q_2$. The high intensity of the peaks suggests a strong diffraction contrast occurs between the central void and the electron density of the structural building unit. This is much like the model Elliot et al. described for the origin of the first sharp diffraction peak observed for inorganic glasses which is based on the spatial persistence of cation-based clusters and voids throughout the structure, analogous to the trimer units and tetrahedral assemblies[56]. It is also useful to note the structural similarities between the tetrahedral assemblies in MIL-100 to the AX$_4$ tetrahedra in many of the AX$_2$ glasses that have been subject to intense debate[23].

Hence, the localised low-$Q$ scattering in MIL-100 partly informs us about the spatial persistence and periodic arrangement of trimer units and tetrahedral assemblies within MIL-100. Of course, it is the delocalised reciprocal-space features that will describe sharp peaks within the PDF and hence explicitly identify the structures of the trimers and tetrahedra based on specific atomic correlations, as discussed later. For completeness, we note that the peak in the calculated structure at 0.23 Å$^{-1}$, which we were unable to measure experimentally, has a real-space period of 27.3 Å which relates to the density fluctuations associated with the pore network of MIL-100[52].

In Fe-BTC, there was an appreciable similarity in the form of the low-$Q$ scattering compared to MIL-100. Furthermore, the peak broadening largely arises from intrinsic sample effects and hence an approximate correlation length can be determined. Fe-BTC contained a peak at 0.75 Å$^{-1}$ (real-space period of 8.4 Å and correlation length of 25.1 Å), accompanied by a weaker peak at 0.42 Å$^{-1}$ (real-space period of 15.0 Å and correlation length of 27.7 Å where FWHM was determined from a symmetrised peak obtained via reflection about the apex), similar in positions to $Q_2$ and $Q_1$, respectively. These correlation lengths are only approximate due to the convolution of the instrumental broadening to the overall peak width and the potential overlap of features. The weak scattering at 0.42 Å$^{-1}$ may be indicative of reduced tetrahedral assembly in Fe-BTC, compared to MIL-100, consistent with our previously reported model[13]. Both peaks are less intense than in MIL-100, suggesting a lower scattering contrast caused by the absence of periodic ordering of the trimer units resulting in a substantially reduced oscillation of electron density propagating through the structure. In other words, the

electron density fluctuation associated with the trimer unit does not vary periodically—as expected from a disordered material.

In $a_m$MIL-100 and $a_m$Fe-BTC there was significant disruption to the low-$Q$ scattering. Both featured a very broad peak, intermediate to $Q_1$ and $Q_2$, centred around 0.63 Å$^{-1}$ (real-space period 10 Å) and 0.65 Å$^{-1}$ (real-space period 9.7 Å), respectively. This suggests a far less uniform distribution of electron density throughout the structures, consistent with their collapsed nature[35].

It is often more common to analyse the real-space PDF form of the scattering data. Nonetheless, it is worthwhile noting the impact of the low-$Q$ scattering on the PDF itself. The contribution of small-angle scattering in the PDF of nanostructured materials is well described and results in a subtle modulation of the PDF baseline[57]. The lower-$Q$ scattering shown in Fig. 3b has a similar effect, describing the density fluctuations of the trimers and tetrahedra. While the exact relationship between the low-$Q$ scattering and its effect in real space is non-trivial, a simple Fourier transform of the low-$Q$ data can be instructive in illustrating how it modulates the baseline of the PDF [Supplementary Fig. 5]. These real space functions highlight the importance of examining the scattering data in both real and reciprocal space, as each representation accentuates different aspects of the data.

**Pair distribution functions**. The four PDFs display broad similarity due to their shared local structure (Fig. 3c). The first three major peaks at $r = 1.98, 3.34$ and $4.68$ Å are largely dominated by the (a) Fe–O, (b) Fe–O and Fe–Fe, and (c) Fe–O, Fe–C and C–O correlations, respectively[13]. More generally, the stronger peaks between 0 and 7 Å largely originate from interactions within the trimer unit and those between 7 and 12 Å arise from the assembly of the trimer units into tetrahedral assemblies (Fig. 1e, f). Beyond 12 Å, peaks reflect the spatial arrangement of tetrahedral assemblies. The subtle modulation of the baseline of the PDF, arising from the low-$Q$ scattering, was apparent in MIL-100 and to some degree in Fe-BTC.

All four materials were similar in the trimer region, whereas greater differences were observed in the tetrahedral region due to the greater proportion of hierarchical assembly occurring in MIL-100 compared to the other materials. Beyond 12 Å, MIL-100 exhibits several large peaks arising from its periodic structure. Fe-BTC, $a_m$MIL-100 and $a_m$Fe-BTC had significantly fewer features present in this region due to their non-crystalline structures.

We find there was a greater difference in the PDFs between the materials pre-amorphisation (MIL-100 and Fe-BTC, a crystalline and a disordered framework, respectively) versus the two materials post-amorphisation ($a_m$MIL-100 and $a_m$Fe-BTC, both amorphous frameworks) [Supplementary Fig. 6]. As a simple measure of this similarity, the Pearson product-moment correlation coefficient between the PDFs was calculated which takes a value between −1 (complete anti-correlation), 0 (no correlation) and +1 (complete correlation)[58,59]. The Pearson's correlation between MIL-100 and Fe-BTC was +0.77, while for $a_m$MIL-100 and $a_m$Fe-BTC it was +0.95, confirming a greater degree of statistical similarity between the amorphous materials (Supplementary Tables 2 and 3). Ultimately, point-wise metrics, such as Pearson's correlation value, consider the PDF as a whole, reducing potentially complex variations in the PDFs to a single value.

Often, in the case of disordered materials, the local structure remains very similar, and it is the extended structure that exhibits greater distortion as local variations are magnified over longer distances; yet it is the local structure to which the PDF is most sensitive. These opposing effects at different length scales make assessing the extent of disorder from the PDF challenging. Simple statistical measures can capture very broad trends between materials with different degrees of disorder. Instead, we sought to capture the subtleties associated with the structural distortion using a PCA approach.

**Comparison between MIL-100 & Fe-BTC**. We begin with a simple comparison between MIL-100 and Fe-BTC. The two experimental PDFs were used as the input functions for PCA, from which two statistically significant principal components were extracted (Fig. 4a). The first principal component accounted for 93.53% of the statistical variation present in the experimental data and bears resemblance to the typical form of a PDF, describing common atom–atom correlations within the data (Fig. 4b). The second principal component captured a much smaller 6.47% of the statistical variance and does not exhibit the typical profile of a PDF, instead, it is a distortion that describes the differences between the two materials (Fig. 4c).

Extending this interpretation, we consider how the percentage of statistical variance captured by these principal components may be used as a simple metric of disorder. That is, we can relate the proportion of variance captured by the distortion component to the degree of disorder in the framework. Of course, this is not attributed as an absolute measure of disorder. Instead, they are considered as relative markers that enable us to flag how the degree of disorder varies between the materials.

**Comparison between $a_m$MIL-100 & $a_m$Fe-BTC**. The same analysis was performed on $a_m$MIL-100 and $a_m$Fe-BTC to examine the similarity between the two amorphous materials. The experimental PDFs were used as the input for PCA, and two significant components were extracted (Fig. 4d). The first principal component accounted for 97.34% of the statistical variation and the second for only 2.66% (Fig. 4e, f). The first component contained atom–atom correlations, while the second described a distortion, similar to the analysis of MIL-100 and Fe-BTC. This suggests the statistical variance captured by the distortion, our simple metric of relative disorder, between $a_m$MIL-100 and $a_m$Fe-BTC is almost two and a half times less than between MIL-100 and Fe-BTC. In other words, the ball-milled materials are more similar to one another than those pre-milling.

Comparing the form of the distortions themselves, the distortion between MIL-100 and Fe-BTC occurs over the entire length scale of the PDF, with considerable distortion at high-$r$. Whereas the distortion between $a_m$MIL-100 and $a_m$Fe-BTC is largely featureless at high-$r$. This suggests that the amorphous materials are largely differentiated by small differences in their local structure. Hence, PCA confirms what we find structurally intuitive: the two amorphous materials ($a_m$MIL-100 and $a_m$Fe-BTC) are *more* similar than the crystalline and disordered materials (MIL-100 and Fe-BTC), with the distortion in the latter pair occurring largely at high-$r$. These broad trends are visually apparent in the data themselves and are also consistent with the Pearson correlations. However, by performing PCA and obtaining a distortion component, we can visualise where the differences in the variance manifest within the PDFs.

An important implication of these results is that the extent of structural collapse occurring between MIL-100 and $a_m$MIL-100 appears greater than for Fe-BTC to $a_m$Fe-BTC. Again, this may have been anticipated given the former transition is from crystalline to an amorphous state, while the latter is from an already disordered state. Nonetheless, we sought to explicitly capture this behaviour and gain insight into the structural collapse of MIL-100 and Fe-BTC to form their amorphous counterparts.

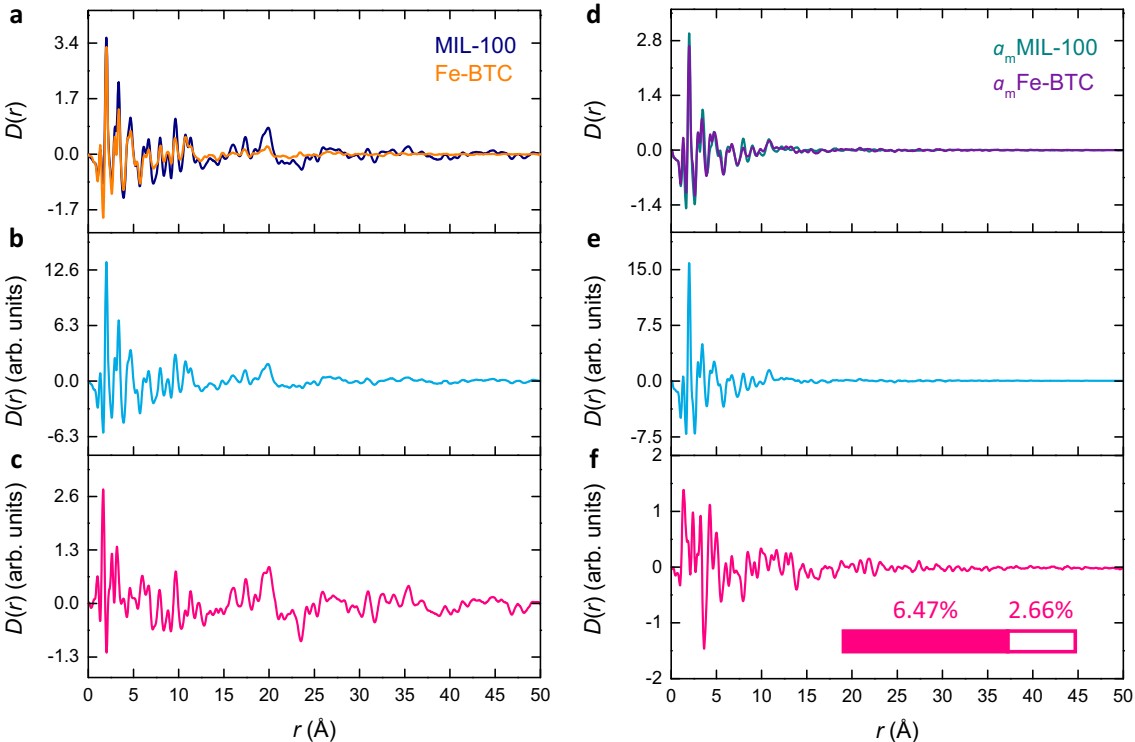

**Fig. 4 Fingerprinting disorder. a** Experimental PDFs for MIL-100 (navy) and Fe-BTC (orange), and the corresponding (**b**) first and (**c**) second principal components extracted through PCA, which describe atom–atom correlations and a structural distortion, respectively. **d** Experimental PDFs for $a_m$MIL-100 (turquoise) and $a_m$Fe-BTC (purple), and the corresponding (**e**) first and (**f**) second principal components extracted through PCA, which describe atom–atom correlations and a structural distortion, respectively. The inset in (**f**) represents the relative magnitudes of the distortion between MIL-100 and Fe-BTC (filled) and $a_m$MIL-100 and $a_m$Fe-BTC (empty).

**Collapse of MIL-100 & Fe-BTC.** Directly studying the mechanism of structural collapse, the way in which the bonds and coordination modes deform under applied stress for example, is no easy undertaking[60,61]. While the precise atomic displacements that occur during collapse will be complex, the PDF will be sensitive to these structural changes and will be captured in the distortion components. These signatures can be used to infer how the local and extended structures are changing.

A series of five ex situ PDFs, taken from Ref. [35], were used to probe the structural collapse of MIL-100 to form $a_m$MIL-100 over the course of 0, 5, 10, 15, and 30 min (Fig. 5a, Supplementary Fig. 7 and Supplementary Table 4). From these, two statistically significant principal components were extracted accounting for 92.36% and 7.59% of the variance, respectively (Fig. 5b, c). The first contained atom–atom correlations, while the latter was a distortion.

As a simple test of the complexity of the distortion component, a dataset was synthesised by sequentially translating the experimental MIL-100 PDF horizontally based on the slight shift in peak position observed in the low-$r$ region (Supplementary Fig. 8). This dataset clearly possessed more long-range correlations than the experimental data. PCA performed on this dataset revealed a distortion component that was very different to the experimentally obtained distortion across all length scales (Supplementary Fig. 9) . Naturally, the real distortion occurring during the collapse of MIL-100 will have a more complex effect than a simple translation of the PDF. Indeed, it was not possible to replicate the experimental distortion component by linear transformation of the MIL-100 data alone.

In order to compare to MIL-100, the same time-resolved ex situ study was performed on Fe-BTC using the I15-1 beamline at Diamond Light Source (Oxfordshire, UK). Specifically, Fe-BTC

was ball milled for 0, 5, 10, 15 and 30 min before X-ray total scattering data were collected. Following data correction and subsequent Fourier transform of the data, PCA was carried out on the five PDFs (Fig. 5d, Supplementary Fig. 10 and Supplementary Table 5). The first two principal components accounted for 96.97% and 2.48% of the variance, describing atom–atom correlations and a distortion, respectively (Fig. 5e, f). The variance associated with the distortion components was over three times greater for the MIL-100 series than for Fe-BTC, suggesting the extent of disordering occurring in the crystalline material is greater, as anticipated.

An initial glance suggests that the distortion components are not strikingly dissimilar (Fig. 6a). While interpretation of the distortion component can be challenging, we can draw out some of the salient features. In general, many peaks are in the same positions, but their intensities are higher in MIL-100 supporting the greater extent of distortion that is occurring. In the low-$r$ region, a particularly prominent feature was observed at 1.60 Å in Fe-BTC corresponding to the appreciable decrease in trough depth upon milling, compared to the trough at 1.60 Å in MIL-100, which remains at a similar depth. This change may be ascribed to the steeper gradient of the PDF contribution that arises from the lower-$Q$ scattering; in Fe-BTC the lower-$Q$ contribution is more comparable to the overall PDF intensity and hence this variation is more significant. Two prominent low-$r$ features were observed at 3.28 and 3.65 Å in MIL-100, together these are an example of the peak shift distortion demonstrated in Fig. 1d. The midpoint at *c.a.* 3.46 Å marks the position of the peak in the PDFs that exhibited the positional shift and is dominated by the first peak in the Fe–Fe partial PDF (i.e. the intra-trimer Fe–Fe distance). A similarly noticeable feature with this same motif was observed between 8.70 and 9.62 Å; the peak

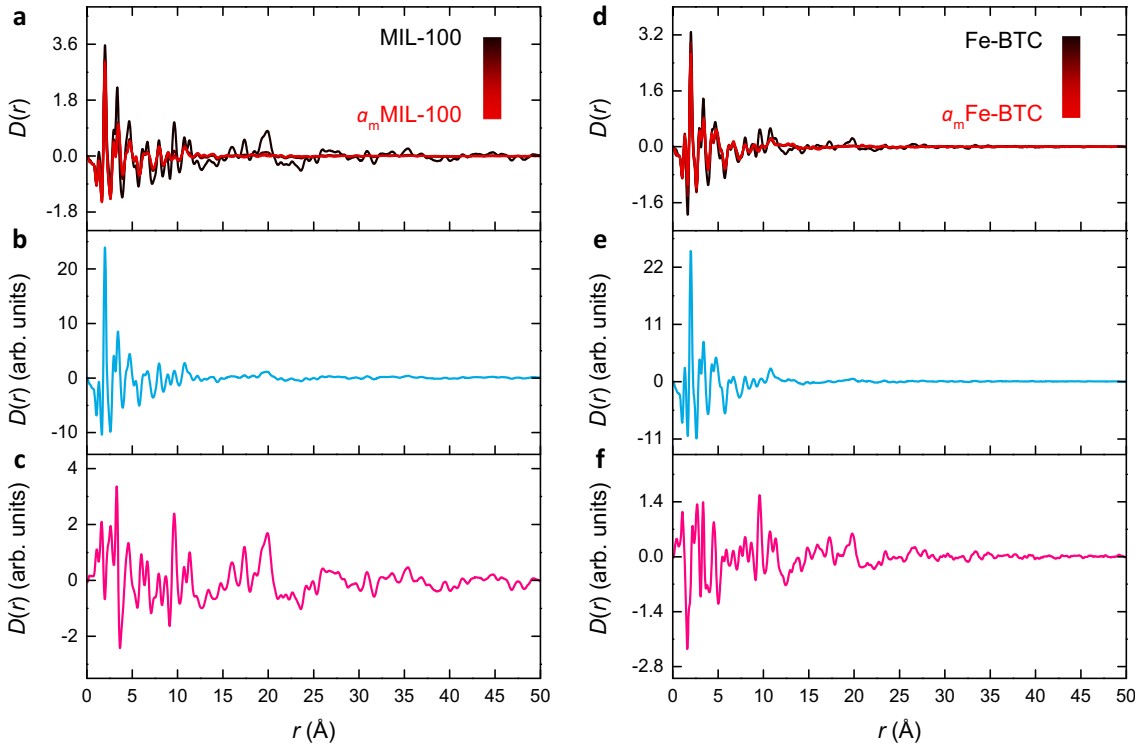

**Fig. 5 Structural collapse. a** Experimental ex situ PDFs following the collapse of MIL-100 over 30 minutes, the corresponding (**b**) first and (**c**) second principal components extracted through PCA, which describe atom–atom correlations and a structural distortion, respectively. **d** Experimental ex situ PDFs following the collapse of Fe-BTC over 30 min, the corresponding (**e**) first and (**f**) second principal components which describe atom–atom correlations and a structural distortion, respectively.

in this region of the PDF is also dominated by the Fe–Fe partial PDF, in this case, the intra-tetrahedral Fe–Fe distance. At high-$r$, as the local distortion propagates through the extended structures, the distortion in the data for MIL-100 contains a considerable number of features associated with the loss of crystallinity. Whereas in Fe-BTC, this region is relatively featureless and flat. Interestingly, the distortion appears to successfully capture the loss of the low-$Q$ scattering in MIL-100 and its contribution to the PDF with a clear fluctuation in the baseline of the distortion above and below zero.

The experimental PDFs can be reconstructed through linear combinations of the principal components. The weightings of these components can be used as a probe of the kinetics of structural collapse, tracking how they vary as a function of time (Fig. 6b). The weightings of the first component remained almost constant throughout both series. The second component, which describes the distortion, exhibited a general decrease with time. We can use the weighting of the distortion and hence the way it augments the underlying structure as a pseudo-reaction coordinate of the structural collapse[62].

The weightings of the distortion varied similarly for both MIL-100 and Fe-BTC, suggesting the rate at which it changes—in effect, the kinetics of the structural collapse—is similar in both materials. The weightings also indicate that the largest change in the distortion occurs within the first five minutes of collapse, apparent from the first derivative of the fitted curves, after which the rate of distortion decreases [Supplementary Fig. 11]. This suggests both materials are most susceptible to collapse in their as-synthesised states, with the progressive introduction of disorder becoming increasingly more challenging.

For clarity, it is important to note that the weightings are discussed in a relative manner. Given no physical restraints are employed within PCA, the raw outputs can seem unintuitive or

non-physical. It is because of this that careful interpretation is required. The first principal component represents over 90% of the variance of the dataset, hence it, in effect, resembles the statistical average of the series and not the end members. It is for this reason that the weighting of the distortion component crosses zero. In some examples where the weightings of the principal components have been used to study in situ reaction kinetics, for example, linear transformations of the components and their weightings can aid interpretation[62]. For example, rescaling the weights arbitrarily can help eliminate confusion from the absolute values themselves and focus on the trends instead. Alternatively, linear regression can be performed using linear combinations of the principal components against the experimental data, which forms the basis of principal component regression techniques[63]. This approach is typically effective when the system contains multiple phases and hence correlations from each of the different phases may become encoded in the same principal component. In the present case, with a single phase undergoing a distortion, such transformations do not provide additional clarity.

If instead, we replace the first principal component with the PDF for MIL-100, new weightings can be obtained via multiple linear regression that are possibly more intuitive (in that they do not cross zero). Nonetheless, the underlying qualitative nature of the weightings and hence the implied transition remains the same (Supplementary Fig. 12). This level of manipulation does however diminish the unbiased and mathematically rigorous nature of pure PCA results. For completeness, preliminary investigations were carried out on the data using NMF. While NMF has been praised for its utility when studying phase mixtures, due to its non-negative constraint, we found that for studying the distortion of a single phase, such as during amorphisation, the results were not as instructive as our current approach.

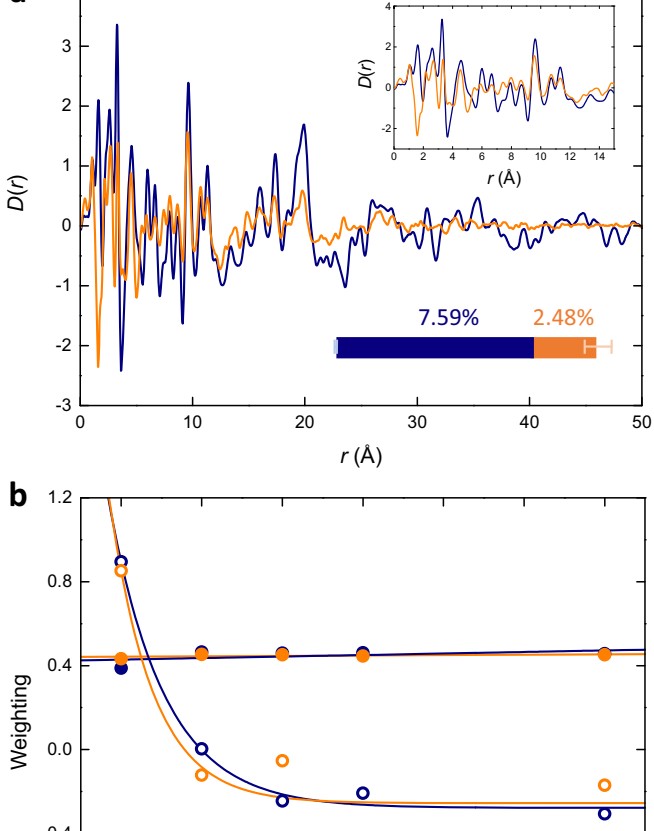

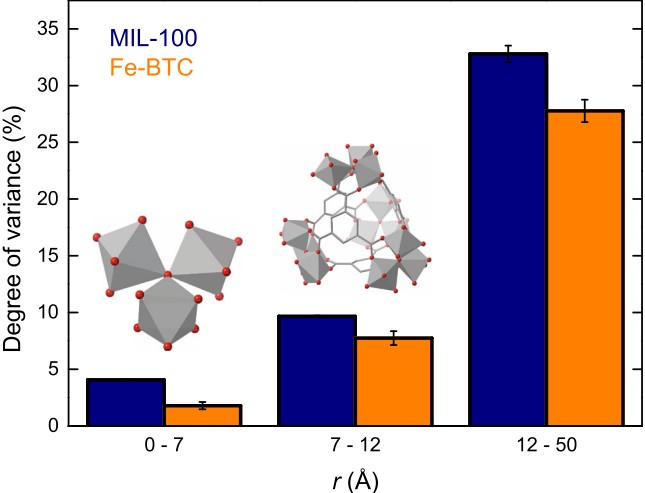

**Fig. 7 Hierarchical structural analysis.** The degree of variance captured by the distortion principal components in MIL-100 (navy) and Fe-BTC (orange) upon structural collapse to form $a_m$MIL-100 and $a_m$Fe-BTC, respectively, in the 0–7 Å (trimer unit), 7–12 Å (tetrahedral assembly), and 12–50 Å (extended structure) regions. Uncertainties represent the variance associated with the discarded principal components. Inset figures show the structure of the trimer unit (left) and tetrahedral assembly (right).

**Fig. 6 Kinetic insight. a** Comparison of the second principal components (distortions) obtained from the structural collapse of MIL-100 (navy) and Fe-BTC (orange). Upper inset shows the distortion at low-$r$. Lower inset represents the relative magnitudes of the distortion occurring upon collapse of MIL-100 (navy) and Fe-BTC (orange), uncertainties represent the variance associated with the discarded principal components. **b** Time-resolved weightings of the first (filled) and second (empty) principal components used to reconstruct the experimental PDF data, fitted using a linear and an exponential decay function, respectively.

In summary, using the distortion component we can extract a fingerprint of the structural collapse that occurs in MIL-100 and Fe-BTC, demonstrating a similar local distortion that has a greater impact as it propagates through the extended structure of MIL-100's crystalline framework. Furthermore, the weightings of the principal components provide insight into the kinetics of structural collapse, suggesting that both materials collapse at a similar rate.

Distortions within the local structure are magnified within the extended structure, yet the PDF has greater sensitivity to the local structure. We, therefore, repeated the analysis using a boxcar approach whereby analysis is performed on subsections of the overall PDF to investigate whether the behaviour is different over different length ranges. Here, the 0–7 Å (trimer unit), 7–12 Å (tetrahedral assembly), and 12–50 Å (extended structure) regions of the PDF were analysed separately to delineate the extent to which the hierarchical structures were changing (Fig. 7). The variance captured by the distortion increased with $r$ in both materials. Chemically this is intuitive, with the trimer unit being

dominated by Fe–O interactions, while the much larger tetrahedral assemblies involve additional metal–linker interactions. These minor changes in the local structure are then magnified over long distances within the extended structure.

As anticipated, the distortion occurring in the extended structure is greater for MIL-100 than Fe-BTC. We may have expected the local structures of both materials to be affected more similarly given their similar chemistry and geometry. However, in both the trimer unit and tetrahedral assembly regions, MIL-100 exhibits a greater distortion compared to Fe-BTC. In fact, MIL-100 exhibits almost twice the variance of Fe-BTC in the trimer region. One possible interpretation of this may be that the local structure of Fe-BTC is more resistant to deformation than in MIL-100; its disordered structure protects itself in effect, compared to the more porous MIL-100 where distortions may propagate more easily. This could have important implications for the macroscopic mechanical properties, potentially pointing towards an enhanced shear modulus in Fe-BTC. Mechanical tests on MIL-100 and Fe-BTC may be a fruitful avenue for future experimental endeavours to confirm this hypothesis.

**Melting of TIF-4.** To investigate the generality of our approach amongst disordered materials, and ensure the results presented for MIL-100 and Fe-BTC were not simply fortuitous, we turn to a very different, both chemically and structurally, system of disorder. Namely, the melting of the crystalline zinc imidazolate framework (ZIF), TIF-4 [$Zn(Im)_{1.8}(mbIm)_{0.2}$, Im – imidazolate and mbIm – 5-methylbenzimidazolate], which occurs at *c.a.* 400 °C where it forms a liquid that we denote $a$TIF-4[64]. Computational studies have suggested that the melting of ZIFs occurs on a picosecond timescale through the de-coordination of a linker followed by subsequent re-coordination of another linker in its place[65]. Understanding the microscopic mechanism of melting in this class of materials is of particular importance for the formation of their macroscopic melt-quenched glasses[54,65].

Variable temperature X-ray PDF data for TIF-4 between 25 and 440 °C were taken from Ref. [64]. (Fig. 8a). The PDFs revealed that the local structure remained intact, even in the liquid state. Specifically, the five key peaks between 0 and 6.5 Å which arise

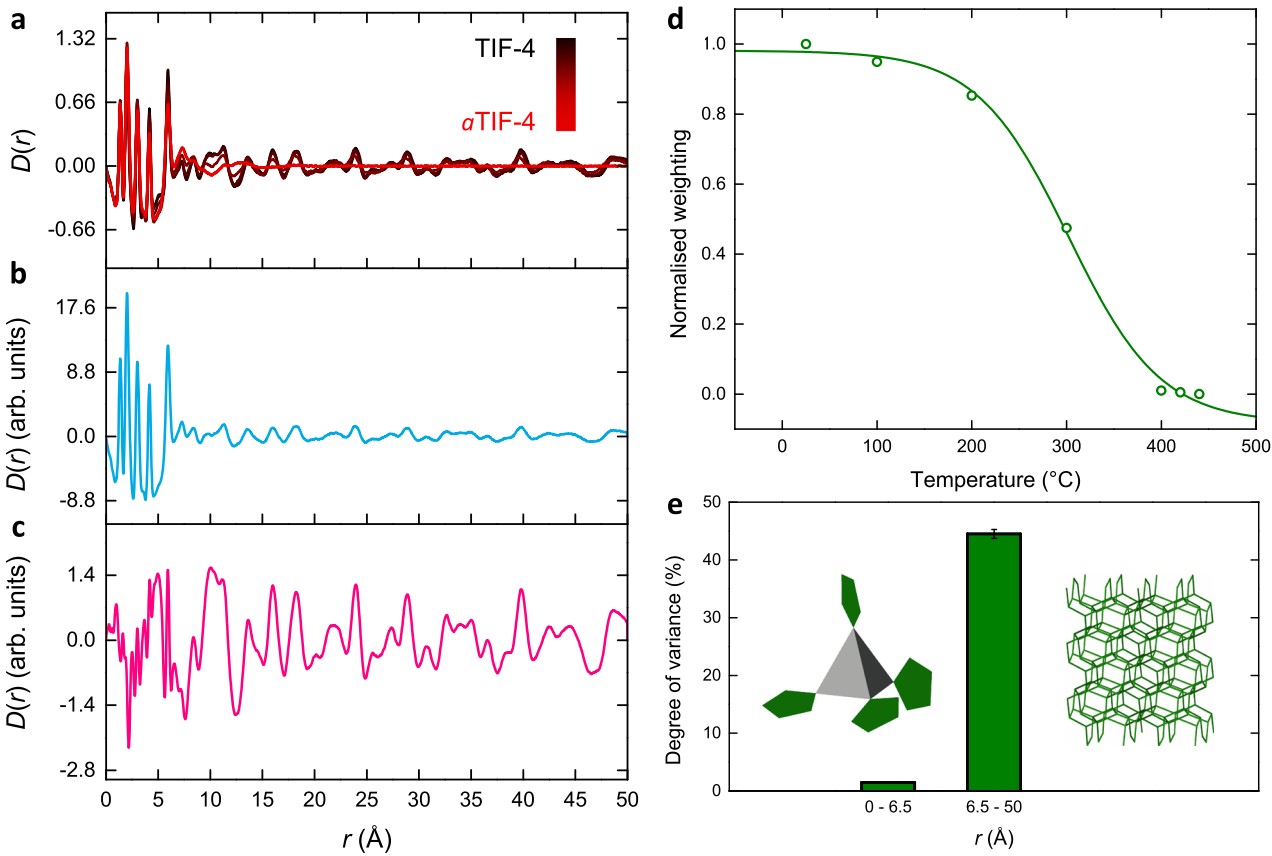

**Fig. 8 Melting of TIF-4. a** Experimental in situ PDFs following the melting of TIF-4, and the corresponding (**b**) first and (**c**) second principal components extracted through PCA, which describe atom–atom correlations and a structural distortion, respectively. **d** Temperature-resolved normalised weightings fitted using a Boltzmann function. **e** The degree of variance captured by the distortion component upon melting of TIF-4, over the length scales of the tetrahedrally coordinated zinc nodes (0–6.5 Å) and the extended structure (6.5–50 Å). Uncertainties represent the variance associated with the discarded principal components. Inset figures show a schematic of the ZnN$_4$ tetrahedra (grey) and imidazolate linkers (green) (left) and the **cag** topology of the extended network structure (right).

from the Zn–Im–Zn structural motif remain present at all temperatures studied. Beyond 6.5 Å, the limit of short-range order, TIF-4 exhibits many peaks even at high-$r$ due to its highly crystalline nature. Upon heating, the intensity of these peaks gradually reduces, with the PDF for $a$TIF-4 entirely featureless in this region.

The same methodology was applied to these TIF-4 data; two statistically significant principal components were extracted using the same criteria as before (Fig. 8b, c). The first component accounted for 93.59% of the variance and the second, a distortion, for 5.86%. Considering the distortion component, we note how different it looks to those obtained for the MIL-100 and Fe-BTC systems which reinforces the idea of the distortion acting as a unique signature of the system and type of disorder. It revealed only subtle changes in the peak profile and baseline in the local structure region of the PDF, consistent with the retention of the Zn–Im–Zn motif. Interestingly, the largest feature in this region of the distortion was at 5 Å which is not commonly a peak position that is described for the Zn–Im–Zn motif[64]. Closer inspection of the partial PDFs reveal that Zn–C and N–C contribute to this region of the PDF, possibly suggesting that these interactions are particularly important during the melting process[66]. Importantly, visual inspection of the PDFs themselves only display a slight reduction in intensity of a shoulder peak at 5 Å which could easily be overlooked; through analysing the distortion component, we uncover the importance of this region during melting. Within the extended structure of the PDF, the

distortion contained more significant features, consistent with the transition from a solid crystal to a liquid.

The weightings of the two components were then determined as a function of temperature (Supplementary Fig. 13). Similar to before, the first component resembles the average, not the end members, of the series and hence the weighting of the distortion crosses zero. In this example, we consider how the weighting of the distortion can be transformed to better represent a reaction coordinate (Fig. 8d). The normalisation of the weightings was performed via a linear transformation to rescale them to the unit interval, as described in Ref. [62], which eliminates the sign ambiguity whilst retaining the shape of the weightings' distribution. The normalised weightings remain largely constant up to 200 °C. A large change then occurs between 200 and 400 °C, after which it remains constant again. This pattern of variation is chemically intuitive; at low temperatures, TIF-4 remains in its solid, crystalline form. Then, as the melting temperature is approached and the melting phenomenon begins to take place, the distortion begins to augment the atomic structure. Once in the liquid state, the distortion remains constant once again. This delayed onset of variation in the distortion component weighting is in direct contrast to the weighting of the distortion when ball milling, which shows large variation almost instantly.

As noted in the distortion component itself, there are only subtle variations below 6.5 Å, which are then magnified within the extended structure. To delineate how structural distortion

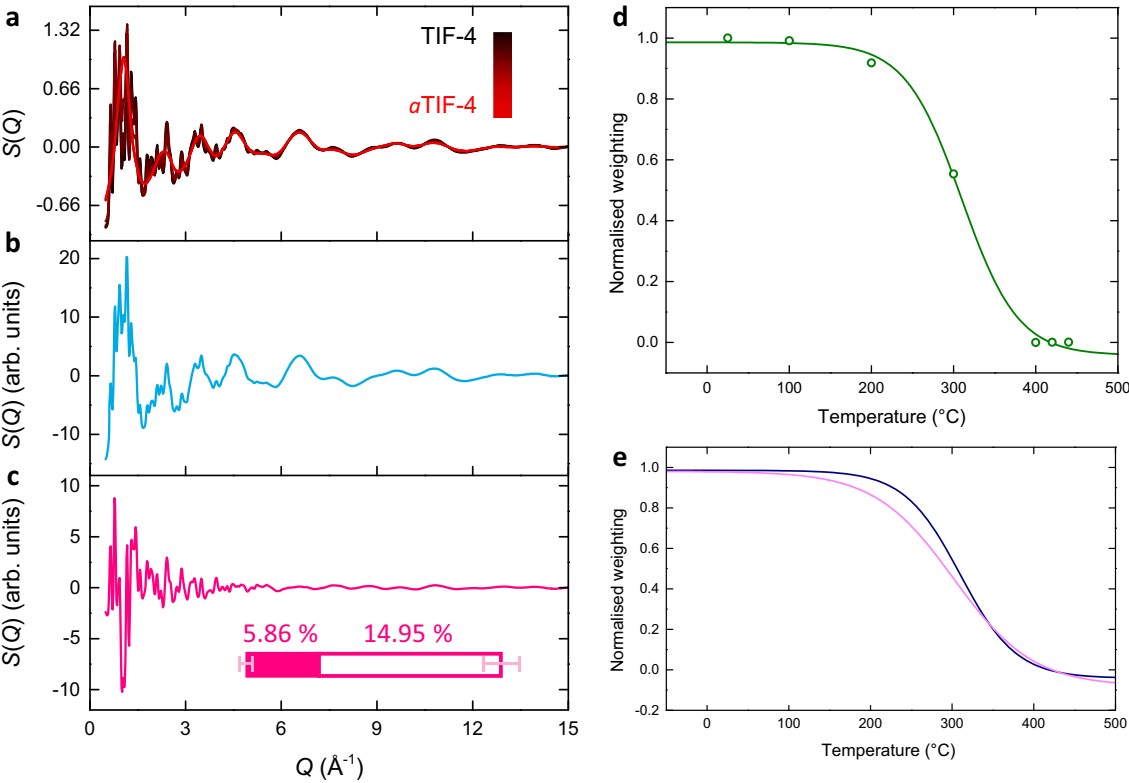

**Fig. 9 Reciprocal space. a** Experimental in situ structure factors following the melting of TIF-4, and the corresponding (**b**) first and (**c**) second components extracted through PCA. Inset shows a comparison of the relative magnitudes of the variance captured by the distortion component in real (filled) and reciprocal (empty) space, uncertainties represent the variance associated with the discarded principal components. **d** Temperature-resolved normalised weightings fitted using a Boltzmann function. **e** Comparison between the Boltzmann functions obtained for the real and reciprocal space weightings of the distortion.

propagates, analysis was carried out on the local and extended structure of TIF-4 using the boxcar approach (Fig. 8e). There was almost a negligible variance in the local structure (<1.5%), compared to a significant degree, over 40%, within the extended structure. The higher variance observed within the local structure of MIL-100 and Fe-BTC is likely due to the greater complexity and size of the building units compared to that in TIF-4. While the higher variance in the extended structure of TIF-4 is attributed to its more crystalline nature, as evidenced by clear peaks in the PDF beyond 40 Å, compared to MIL-100 and Fe-BTC.

As a further proof of concept, we explored how this approach worked in reciprocal space. The experimental structure factors demonstrated a clear crystalline–amorphous transition, with Bragg peak intensity reducing with temperature and only diffuse scattering being observed for aTIF-4 (Fig. 9a). The first and second principal components of the structure factors, along with their weightings, were then extracted and transformed as before (Fig. 9b, d and Supplementary Fig. 14). Crucially, we find excellent agreement between the weightings obtained in both real and reciprocal space (Fig. 9e). It is interesting to note there is almost a threefold increase in the variance associated with the distortion in reciprocal space than in real space. This arises from the greater sensitivity of the real and reciprocal space data to the local and average structure, respectively, coupled with the amplification of minor changes in the local structure within the extended structure as the distortion propagates throughout the framework.

As a final evaluation of our approach, the Fourier transform of the distortion obtained in reciprocal space was calculated to determine its effect in real space. We find there is excellent agreement between this Fourier transform and the distortion

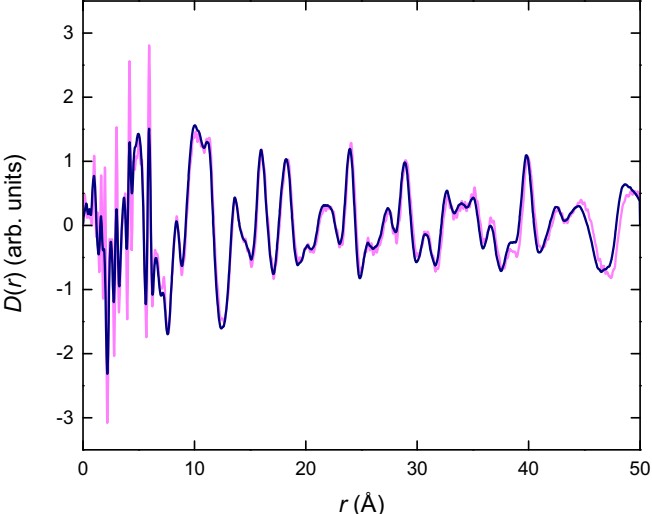

**Fig. 10 Fourier transformation.** Comparison between the distortion component obtained in real space (navy) and the Fourier transform of the distortion component obtained in reciprocal space (pink).

directly obtained in real space (Fig. 10). Not only does this confirm the self-consistency of the analysis performed here, but more importantly it validates the notion that these fingerprints of disorder are intrinsically encoded within the PDF. Hence, we can foresee this approach as a powerful route for which these mechanisms can be studied in the future, in either real or reciprocal space.

## Discussion

In this work, we have demonstrated how PCA can be applied to the study of structural disorder—focusing in particular on the study of structural distortions. First, we investigated the mechanical collapse of MIL-100 and Fe-BTC, probing the extent and rate of disordering that occurs upon ball milling. The experimental PDFs were decomposed into their principal components, thereby enabling us to study the motifs of disorder directly through the distortion components. We find the distortion component to be a unique fingerprint of the structural mechanism of disorder, while the degree of variance can be used as a simple metric of relative disorder. The weightings of the components were used to provide kinetic insight. We find the rate at which the two materials collapse is similar, yet the magnitude of distortion that occurs in the crystalline MIL-100 series is greater.

As proof of the potential generality of this methodology, we studied the melting of TIF-4 in both real and reciprocal space, obtaining results consistent with our understanding of the melting process. Hence, we were able to obtain chemically meaningful and intuitive insight into two very distinct chemical systems, ball milling-induced amorphisation of iron–carboxylate frameworks and melting of a zinc–imidazolate framework.

While the use of PCA has been applied across a spectrum of fields, the focus within PDF studies has largely been multi-phase systems where atom–atom correlations have received the most attention. The power of PCA as a tool to study processes of disorder has been widely underappreciated. We have demonstrated that studying distortions can provide powerful mechanistic and kinetic insight into these processes that are notoriously challenging to characterise. PCA is a promising route to readily capture, quantify, and fingerprint these motifs of disorder. Furthermore, given the direct interplay between structure and property in MOFs, it is likely that our structural insights will also correlate with the physical properties of these systems (e.g. surface area or pore size). We envisage that this approach has generality towards a whole host of systems including disorder introduced through (i) the application of heat, pressure, or radiation[54,67,68], (ii) de novo syntheses of missing linker or node defects[69,70], and (iii) compositional inhomogeneity of mixed-linker or mixed-metal systems[10,71]. Many of these disordered materials have already been exploited as proton conductors, broadband dielectrics, and battery materials, yet our fundamental understanding of their structural disorder remains poor compared to the vast majority of crystalline materials[72–74].

Disorder is no longer a phenomenon to be avoided. As we discover that it is not only present in many supposedly crystalline frameworks, but also underpins much of the functionality in this rich class of materials, it is crucial that we are able to characterise its structural impact to take full advantage of its utility.

## Methods

**Synthesis**. All chemicals were used as received. Iron (III) nitrate nonahydrate (99.95%), 1,3,5-benzenetricarboxylic acid (95%), methanol (99.8%), ethanol (99.8%), ammonium fluoride (99.99%), sodium hydroxide pellets (98%) and iron (II) chloride tetrahydrate (99.99%) were all purchased from Sigma Aldrich.

**MIL-100**. MIL-100, $Fe_3O(OH)(H_2O)_2[(C_6H_3)(CO_2)_3]_2 \cdot nH_2O$, was synthesised and purified according to Refs. [75,76], respectively. 1, 3, 5-benzentricarboxylic acid (1.676 g) was dissolved in an aqueous 1 M solution of sodium hydroxide (23.720 g) and added dropwise to a solution of iron (II) chloride tetrahydrate (2.260 g) dissolved in water (97.2 mL). The solution was left to stir for 24 hours at room temperature before being centrifuged, washed with ethanol (3 × 20 mL), and then dried overnight at 60 °C. The powder was dispersed and heated for 3 hours each in water (700 mL at 70 °C), ethanol (700 mL at 65 °C), then aqueous 38 mM ammonium fluoride solution (700 mL at 70 °C). The powder was recovered between each stage by centrifugation. The resultant powder was dried overnight at 60 °C.

**Fe-BTC**. Fe-BTC was synthesised following the procedure in Ref. [13]. Iron (III) nitrate nonahydrate (2.599 g) and 1,3,5-benzentricarboxlic acid (1.177 g) were each dissolved in 20 mL of methanol. The two solutions were combined and left to stir for 24 hours at room temperature, forming a viscous orange liquid. It was then washed with ethanol (3 × 20 mL) and dried overnight at 60 °C. The powder was then purified as described above.

**Ball milling**. Before ball milling, MIL-100 and Fe-BTC were activated at 150 °C under dynamic vacuum overnight. For each duration of milling, 100 mg of activated powder was sealed inside a 10 mL stainless steel jar with one 7 mm diameter stainless steel ball bearing. The jar was placed in a Retsch MM 400 grinder mill and ball milled at 25 Hz for 5, 10, 15, and 30 min.

**Powder X-ray diffraction**. Data were collected at room temperature using a Bruker D8 diffractometer using Cu $K\alpha_1$ ($\lambda = 1.5406$ Å) radiation and a LynxEye position-sensitive detector with Bragg-Brentano parafocusing geometry. Samples of finely ground powder were dispersed onto a low background silicon substrate and loaded onto the rotating stage of the diffractometer. Data were collected over the scattering angle range $5° < 2\theta < 45°$. Pawley refinement was carried out using TOPAS Academic (V6) software[77]. The unit cell parameters were refined against those previously reported for MIL-100 [52].

**X-ray total scattering**. Data were collected at the I15-1 beamline (Diamond Light Source, UK) using synchrotron radiation ($\lambda = 0.161669$ Å, 76.7 keV). Prior to measurement, samples were activated under dynamic vacuum at 150 °C overnight. A small amount of finely ground sample was then loaded into a borosilicate capillary (inner diameter of 1.17 mm) to a height of 3.5 cm. Capillaries were sealed and mounted onto the instrument. Data were collected under ambient conditions for each sample, an empty capillary, and the blank instrument over the region $\sim 0.4 < Q < \sim 26$ Å$^{-1}$. The raw total scattering data were corrected for background, multiple scattering, container scattering and Compton scattering along with absorption corrections, using GudrunX following standard procedures over the range $\sim 0.6 < Q < \sim 23$ Å$^{-1}$ [78]. The Fourier transform was then calculated to obtain the pair distribution function.

**Principal component analysis**. Principal component analysis (PCA) was used to determine the latent variables of the PDF data. PCA is a dimensionality reduction technique that projects the original data onto an orthogonal basis whose directions are found by simultaneously maximizing the variance of the data and minimising the reconstruction error. PCA is widely implemented into many software packages such as MATLAB and RStudio, here we used the PCA function within the Origin Pro software. Firstly, a matrix, $X$, of the independent variables was constructed, standardised to obtain the matrix of Z-scores, $Z$, and the symmetric covariance matrix, $C$, of $Z$ was calculated. Note, the covariance matrix of standardised data is the correlation matrix. The use of the correlation matrix is typically associated with datasets on different scales. Given the challenge of placing the X-ray PDF on an absolute scale, analysis of the correlation matrix was chosen to reduce uncertainties introduced during the normalisation of the total scattering data as it is sensitive to relative changes in the PDF but not linear transformations. The eigenvectors and their corresponding eigenvalues of $C$ were then calculated, such that $C\ V = \lambda\ V$, where $V$ is the matrix of eigenvectors and $\lambda$ is a diagonal matrix of the eigenvalues. The eigenvalues were then sorted from largest to smallest and the eigenvectors sorted accordingly, denoted $V^*$. The principal components, $Z^*$, were then obtained by projecting the original data onto the new orthogonal basis set $V^*$. Components of $Z^*$ were retained such that a minimum threshold of 98% of the variance was captured. In all cases, 98% of the variance was captured by the first two components, except for the boxcar analysis of the extended structure of Fe-BTC and TIF-4 where a second distortion component was required.

## Data availability

The experimental data that support the findings of this study are provided in the manuscript, its supplementary information, or are available at https://doi.org/10.17863/CAM.80890.

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

## Acknowledgements

AFS would like to thank H. S. Geddes and A. L. Goodwin (University of Oxford), C. S. Coates and A. Appios (University of Cambridge), and D. S. Keeble (Diamond Light Source) for useful discussions regarding the multivariate analysis. AFS acknowledges the EPSRC for a PhD studentship under the industrial CASE scheme along with Johnson Matthey PLC (JM11106). AMB acknowledges the Royal Society for funding (RGF\EA \180092) as well as the Cambridge Trust for a Vice Chancellor's Award (304253100). KEJ thanks the Royal Society for a University Research Fellowship and the European Research Council under FP7 (CoMMaD, ERC Grant No. 758370). TDB thanks the Royal Society for a University Research Fellowship (UF150021), the Leverhulme Trust for a Philip Leverhulme Prize, and the University of Canterbury Te Whare Wānanga o Waitaha, New Zealand, for a University of Cambridge Visiting Canterbury Fellowship. We extend our gratitude to Diamond Light Source, Rutherford Appleton Laboratory, U.K., for access to beamline I15-1 (EE20038).

## Author contributions

A.F.S. conceptualised and designed the project. T.D.B. supervised the project. T.D.B. and T.J. acquired funding. A.F.S. synthesised and characterised the samples. A.F.S. and P.A.C. collected the total scattering data. A.F.S. analysed and interpreted the total scattering data with the help of D.A.K. A.M.B. contributed the total scattering data for TIF-4 from Ref. [64]. A.F.S. performed the multivariate analysis. I.B. and K.E.J. contributed useful discussions. A.F.S. wrote the manuscript and all authors contributed to the final version.

## Competing interests

T.J. works for Johnson Matthey PLC, a company with an interest in the commercialisation of MOF materials. The remaining authors declare no competing interests.
