## [Peer Review File · Nature Communications]

Multivariate Analysis of Disorder in Metal–Organic FrameworksREVIEWER COMMENTS

Reviewer #1 (Remarks to the Author):

This work describes the application of principal component analysis (PCA) of pair distribution function data to assess the kinetics and mechanism of structural collapse in disordered metal-organic frameworks. In particular, the authors analyse data of three different systems: MIL-100, Fe-BTC and TIF-4. As the authors highlight in the introduction, "characterising disorder is challenging". Therefore, the development of new approaches to deal with the characterisation of challenging disordered materials is welcome and needed! The opinion of this referee is that this manuscript should be publishable in Nature Communications after minor revision.

Before explaining my minor concerns, I would like to say that I have really enjoyed reading this interesting paper. The article is clear and concise, and explains in detail the PCA methodology applied to PDF analyses in a pedagogical way. So I would like to thank the authors for that.

Here are my comments about this work:

-In my opinion, one of the most remarkable results is shown in Figure 8 (kinetics of the amorphisation of MIL-100 and Fe-BTC). The authors explained several times along the article, that the absolute values of the weightings are not chemically intuitive (although mathematically correct). In Figures 10d, 11d, the weightings have been then normalised and presented as reaction coordinates. Could you expand a bit on the procedure followed to normalise the weightings?

- Another very interesting finding is coming from the hierarchical analyses. The local structure of the topological disordered Fe-BTC seems to be more resistant towards amorphisation (that is, changes) than that of crystalline MIL-100. This is unexpected indeed. Beyond the amorphisation, do the authors envisage that the clusters in crystalline MOFs are more prone to suffer local distortions or even chemical modifications?

- Along the article, the authors applied the PCA analyses to both reciprocal and real spaces, resulting in quite similar results (Figure 12, which is very nice!). Are both approaches equally correct? What are the thoughts of the authors in this regard?

- Could you add which software has been used to carry out the PCA analyses and to compute the partial structure factors?

Other minor changes are:

1. Figure 8 (Page 11), should be placed after the paragraph "An initial glance...":
2. Revise y axis in Figures 10d, 11d, 11e

Reviewer #2 (Remarks to the Author):

The authors of this work have presented a compelling study of disorder in several porous materials using a simple PCA data analysis technique of pair-distribution functions (PDFs) obtained from synchrotron data.

I mostly enjoyed reading the paper and have enclosed a few constructive critiques below / requests for improvement. They are listed below in no particular order:

- I feel that the spirit of the paper is suggesting a new *method* for analyzing / understanding disorder (as opposed to an emphasis on the disorder insights of the particular materials chosen). However, the paper is relatively light on method details. The method is described in relatively broad strokes and light on the details needed to reproduce the work of the authors. I thought perhaps the details would be in the SI but I couldn't find them there. If I wanted to assign a grad student to follow this paper's methods on a set of x-ray scattering data I obtained from a collaborator to get similar data, I think they would have a hard time figuring out how to implement the methods from the details given. If the

authors could expand the PCA section in the method section to be a little more detailed + pedagogical, I think that would help the paper have a greater impact (if it doesn't seem suitable for the main manuscript, then perhaps a page or two in the SI would be great).

--- Was there some standard software package used to do the analysis, or were the authors using their own code? This question is less relevant if the precise implementation details are provided, but in lieu of those the code + input parameters could substitute.

- Somewhat related to the above point, I found parts of the manuscript to be inscrutable, as if written for an audience of structure analysis experts. For example, in the "Hierarchical Analysis" section, the authors write "We therefore repeated the analysis using a boxcar approach...". I have no idea what this means, and I suspect most readers wouldn't know either. There are many other places where the text is similarly hard to parse. What is the " $-4\pi\rho r$ " baseline?

- Could the authors comment on whether the PCA components / signatures could be used to extract some guess of changes in more familiar properties, such as pore size / surface area? It would be interesting to know how disorder affects these more familiar properties, especially as a function of time / ball milling etc.

Overall I think this is high quality work. I just wish it was easier to understand / follow for this lower IQ reviewer :)

REVIEWER COMMENTS

Reviewer #1 (Remarks to the Author):

This work describes the application of principal component analysis (PCA) of pair distribution function data to assess the kinetics and mechanism of structural collapse in disordered metal-organic frameworks. In particular, the authors analyse data of three different systems: MIL-100, Fe-BTC and TIF-4. As the authors highlight in the introduction, “characterising disorder is challenging”. Therefore, the development of new approaches to deal with the characterisation of challenging disordered materials is welcome and needed! The opinion of this referee is that this manuscript should be publishable in Nature Communications after minor revision.

We are glad the reviewer highlighted the importance of our work in helping to understand complex disordered systems and would like to thank them for their time in reading and providing valuable feedback.

Before explaining my minor concerns, I would like to say that I have really enjoyed reading this interesting paper. The article is clear and concise and explains in detail the PCA methodology applied to PDF analyses in a pedagogical way. So, I would like to thank the authors for that.

We very much appreciate these kind comments!

Here are my comments about this work:

-In my opinion, one of the most remarkable results is shown in Figure 8 (kinetics of the amorphisation of MIL-100 and Fe-BTC). The authors explained several times along the article, that the absolute values of the weightings are not chemically intuitive (although mathematically correct). In Figures 10d, 11d, the weightings have been then normalised and presented as reaction coordinates. Could you expand a bit on the procedure followed to normalise the weightings?

We are glad the reviewer enjoyed the kinetic insight obtained from our analysis. Importantly, they have identified one of the more subtle challenges faced in this work. While PCA itself is mathematically rigorous, all physical meaning is given to the results by interpreting them in the context of our chemical understanding. When considering the weightings, it is the relative change in magnitude that is most informative, opposed to their absolute value. In the literature, PCA weightings have been reported in their absolute form, in various normalised forms or in some cases without scale bars. The latter two are used to overcome the slightly unintuitive nature of absolute values.

In our first examples of MIL-100 and Fe-BTC, we reported the weights in their absolute form. This was chosen to reinforce the pedagogical nature of the paper, beginning with raw PCA outputs and then introducing the concept of normalisation later on. In the case of TIF-4, a linear transformation was used (as described in Ref. 62) to rescale the weightings to the unit interval. Because of the linear nature of this transformation, the shape of the weighting's

distribution remains intact (retaining the important kinetic information), while eliminating the sign-confusion. This normalisation is well-suited to processes that have a clear start and end point – the structural transition from the crystalline solid to the liquid in this case. Normalised weightings were obtained via the following transformation:

$$W_i^{\text{norm}} = \frac{W_i - W_{\min}}{W_{\max} - W_{\min}}$$

We have clarified our discussion regarding the normalisation process of the TIF-4 datasets as follows:

The normalisation of the weightings was performed *via* a linear transformation to rescale them to the unit interval, as described in Ref. 62, which eliminates the sign ambiguity whilst retaining the shape of the weightings' distribution.

- Another very interesting finding is coming from the hierarchical analyses. The local structure of the topological disordered Fe-BTC seems to be more resistant towards amorphisation (that is, changes) than that of crystalline MIL-100. This is unexpected indeed. Beyond the amorphisation, do the authors envisage that the clusters in crystalline MOFs are more prone to suffer local distortions or even chemical modifications?

We agree that the implications of the hierarchical analysis are very interesting and is certainly a promising avenue for future studies on the impact of structural disorder on mechanical properties. As the reviewer states, our evidence suggests that the statistical variation, and thus the atomic structure, varies more in MIL-100 than in Fe-BTC across all length scales of the hierarchical structure. Hence, it appears that Fe-BTC is more resistant to amorphisation.

In our opinion, we would not expect that the clusters within crystalline MOFs are inherently much different from those in the disordered structures. Instead, the well-ordered structure of the crystalline framework is perhaps more susceptible to propagating structural distortions. Similarly, we would not anticipate the inherent chemical reactivity of the clusters in crystalline MOFs to be much different from those in the disordered MOFs (ignoring the potential for greater numbers of defect sites). However, the clusters within a crystalline framework are likely to be more accessible than in a disordered structure, which may give rise to a potential increase in their chemical reactivities. In other words, the presence of extended structure likely impacts the physical and chemical properties of the frameworks, more so than the local structure itself.

We have clarified this in the manuscript as follows:

One possible interpretation of this may be that the local structure of Fe-BTC is more resistant to deformation than in MIL-100; its disordered structure protects itself in effect, compared to the more porous MIL-100 where distortions may propagate more easily.

- Along the article, the authors applied the PCA analyses to both reciprocal and real spaces, resulting in quite similar results (Figure 12, which is very nice!). Are both approaches equally correct? What are the thoughts of the authors in this regard?

The reviewer raises an interesting point here and it was something that we considered during our analysis. The PDF (real space) and structure factor (reciprocal space) are two ways of representing total scattering data and are linked *via* the Fourier transform. While the same total scattering data is used to obtain both the real and reciprocal space, the two forms have different emphases. The real space data are more sensitive towards local structure (the largest proportion of the intensity is at lower- r). While the reciprocal space data are more sensitive towards extended structure (the largest proportion of the intensity is at lower- Q).

We initially began our analysis in real space and once we had found our results to be chemically meaningful, we considered whether the analysis would also work in reciprocal space. Performing the analysis in reciprocal space yielded weightings that matched the real-space results almost perfectly. This established that the same kinetic insight was encoded within both datasets. To check whether the distortion component itself was the same in both real and reciprocal space, the Fourier transform was calculated of the latter. This revealed almost identical results to those obtained directly in real space (Fig. 12). Hence, the identical results can be obtained in real or reciprocal space, and both are equally valid.

The key difference between the two routes arises in the relative magnitudes of the variance captured by the distortion component (Fig. 11c inset). The relative size of the distortion in reciprocal space is three times that in real space. This effectively captures the different sensitivity of the real and reciprocal forms of the data. Upon melting TIF-4, small distortions occur in the local structure which then propagate into large distortions in the extended structure. The PDF is more sensitive to the small local distortions and hence the degree of variance is much smaller.

Mathematically, it is reasonable to perform PCA in either real or reciprocal space. In practice, we anticipate this analysis will commonly be carried out in real-space due to its more intuitive nature. We have briefly added a comment to the manuscript highlighting that this approach can successfully be used in either real or reciprocal space.

Hence, we can foresee this approach as a powerful route for which these mechanisms can be studied in the future, in either real or reciprocal space.

- Could you add which software has been used to carry out the PCA analyses and to compute the partial structure factors?

We have added both pieces of software to the relevant sections.

Other minor changes are:

1. Figure 8 (Page 11), should be placed after the paragraph "An initial glance...":

Figure 8 has now been moved.

2. Revise y axis in Figures 10d, 11d, 11e

The y-axes on these figures now read “Normalised weighting”, in line with the discussion above.

Reviewer #2 (remarks to the author)

The authors of this work have presented a compelling study of disorder in several porous materials using a simple PCA data analysis technique of pair-distribution functions (PDFs) obtained from synchrotron data.

I mostly enjoyed reading the paper and have enclosed a few constructive critiques below / requests for improvement. They are listed below in no particular order:

We are glad the reviewer enjoyed the manuscript and would like to thank them for their time in reading and providing valuable feedback.

- I feel that the spirit of the paper is suggesting a new *method* for analyzing / understanding disorder (as opposed to an emphasis on the disorder insights of the particular materials chosen).

We are pleased that our use of two very different systems in this study clearly communicated the generality of this approach.

However, the paper is relatively light on method details. The method is described in relatively broad strokes and light on the details needed to reproduce the work of the authors. I thought perhaps the details would be in the SI but I couldn't find them there. If I wanted to assign a grad student to follow this paper's methods on a set of x-ray scattering data I obtained from a collaborator to get similar data, I think they would have a hard time figuring out how to implement the methods from the details given. If the authors could expand the PCA section in the method section to be a little more detailed + pedagogical, I think that would help the paper have a greater impact (if it doesn't seem suitable for the main manuscript, then perhaps a page or two in the SI would be great).

--- Was there some standard software package used to do the analysis, or were the authors using their own code? This question is less relevant if the precise implementation details are provided, but in lieu of those the code + input parameters could substitute.

We'd like to thank the reviewer for this honest comment and appreciate that the manuscript would benefit from additional details. Indeed, we spent significant time and effort understanding the PCA method ourselves before carrying out this work.

PCA is widely implemented as a “black-box” solution in many software packages such as MATLAB and RStudio. Our original intention was to provide a detailed theoretical background so that the reader can understand how the PCA method is carried out by these packages – where PCA is often a “one-click” process. However, omitting the practical details was clearly an oversight and we have now included them.

PCA is widely implemented into many software packages such as MATLAB and RStudio, here we used the PCA function within the Origin Pro software.

- Somewhat related to the above point, I found parts of the manuscript to be inscrutable, as if written for an audience of structure analysis experts. For example, in the "Hierarchical Analysis" section, the authors write "We therefore repeated the analysis using a boxcar approach...". I have no idea what this means, and I suspect most readers wouldn't know either. There are many other places where the text is similarly hard to parse. What is the " $4\pi\rho r$ " baseline?

We regret not keeping the reader in mind when making these statements and agree that the manuscript would benefit from extra clarity in these more technical areas of the analysis. To that end, we have revisited the areas highlighted by the reviewer to ensure the work can be enjoyed by a wide readership. Several grammatical errors in the manuscript were also corrected in the process.

For PDFs obtained from X-ray scattering data, this takes the form of positive peaks above a $-4\pi\rho r$ baseline (which is most obvious at low- r , ρ is the sample density) and apparent envelope to the PDF peaks that decreases at high- r .¹⁸

We, therefore, repeated the analysis using a boxcar approach whereby analysis is performed on subsections of the overall PDF to investigate whether the behaviour is different over different length ranges. Here, the 0 to 7 Å (trimer unit), 7 to 12 Å (tetrahedral assembly), and 12 to 50 Å (extended structure) regions of the PDF were analysed separately to delineate the extent to which the hierarchical structures were changing [Fig. 9].

- Could the authors comment on whether the PCA components / signatures could be used to extract some guess of changes in more familiar properties, such as pore size / surface area? It would be interesting to know how disorder affects these more familiar properties, especially as a function of time / ball milling etc.

This is an excellent suggestion and one that we are actively pursuing. From the analysis performed on Gaussian functions (Fig. S1), we observed how simple peak translations / broadening manifest in the distortion component. The experimental distortions proved significantly more complex than these, though we were able to identify certain motifs within them.

In the manuscript, we have demonstrated that our results possess meaningful structural insight. Given the intrinsic relationship between structure and properties within MOFs, we are confident there will be a significant interplay between our results and the physical properties of these systems. In order to bridge this gap in our understanding, we have investigated the correlation between the weightings obtained via PCA and the BET surface area in the MIL-100 series below.

In Reviewer Figure 1, we have compared the weightings of the distortion in the MIL-100 series to our previously published data on the relative BET surface areas of these materials. Both datasets exhibit a general decrease with milling time. However, the rate at which the surface

area decreases is greater than the distortion. We anticipate it may be possible to develop empirical relationships between PCA analysis and physical properties through additional experimental studies.

Reviewer Figure 1 Comparison between the normalised weighting of the distortion component from the MIL-100 series (black) and the corresponding proportion of retained BET surface area (red). Gas sorption data from Sapnik *et al.*, *Dalton Trans.*, (2021).

We have updated our discussion to include this suggestion and hope that it will provoke further interest into the wide applicability of our methodology.

Furthermore, given the direct interplay between structure and property in MOFs, it is likely that our structural insights will also correlate with the physical properties of these systems (e.g. surface area or pore size).

Overall I think this is high quality work. I just wish it was easier to understand / follow for this lower IQ reviewer :)

We would like to thank the reviewer again for their positive feedback. We believe that the amendments made not only make this manuscript suitable for publication in Nature Communications but also a valuable reference for researchers looking to implement this method in their own studies.

REVIEWER COMMENTS

Reviewer #1 (Remarks to the Author):

The authors have satisfactorily addressed all my concerns and comments.
Very nice work - in my opinion, with a lot of potential for becoming a highly cited paper.

Reviewer #2 (Remarks to the Author):

Thank you for making revisions to this work. I have no further comments and commend the authors on a great paper.